# Association between Novel Hematological Indices and Measures of Disease Activity in Patients with Rheumatoid Arthritis

**DOI:** 10.3390/medicina59010117

**Published:** 2023-01-06

**Authors:** Jung-Yoon Choe, Chan Uk Lee, Seong-Kyu Kim

**Affiliations:** 1Division of Rheumatology, Department of Internal Medicine, Catholic University of Daegu School of Medicine, Daegu 42472, Republic of Korea; 2Department of Internal Medicine, Kwak’s Hospital, Daegu 41919, Republic of Korea

**Keywords:** rheumatoid arthritis, hematological index, composite measure, disease activity

## Abstract

*Background and Objectives*: Hematological indices have been known to be available markers used for evaluating disease activity in rheumatoid arthritis (RA). This study serves to verify the association between four different hematological indices and disease activity measures in patients with RA. *Materials and Methods*: The study included 257 female RA patients and 71 age-matched female controls. Four hematological indices, namely systemic immune-inflammation index (SII), neutrophil-to-hemoglobin and lymphocyte (NHL) score, neutrophil-to-lymphocyte ratio (NLR), and platelet-to-lymphocyte ratio (PLR), were evaluated. Composite measures of RA included Disease Activity Score 28 joints (DAS28), the simplified disease activity index (SDAI), and the clinical disease activity index (CDAI). *Results*: Patients with RA showed statistically higher SII, NHL score, NLR, and PLR compared with controls. SII and NHL score were significantly associated with DAS28 erythrocyte sedimentation rate (DAS28-ESR), DAS28 C-reactive protein (DAS28-CRP), CDAI, and SDAI, whereas NLR was related to DAS28-CRP, CDAI, and SDAI. SII, NHL score, and NLR tended to increase as disease activity based on DAS28-ESR, DAS28-CRP, and CDAI worsened. In the analysis using receiver operating characteristic curve of hematological indices for diagnostic accuracy, the area under the curve was 0.703 (95% confidence interval, CI 0.637–0.769, *p* < 0.001) for SII and 0.705 (95% CI 0.639–0.770, *p* < 0.001) for NHL score, which showed acceptable potential for the diagnosis of RA. Four hematological indices showed weak potential for the detection of remission. *Conclusions*: The present study results showed that SII and NHL scores might be useful markers that adequately reflect disease activity and lead to more accurate diagnosis in RA.

## 1. Introduction

Rheumatoid arthritis (RA) is a chronic systemic inflammatory disease characterized by synovial inflammation, bone erosion, and cartilage destruction that lead to joint damage and deformity, with a worldwide prevalence of approximately 1% of the general population [1,2]. Furthermore, RA causes increased risk of disability, mortality, and morbidity. Recent advanced understanding of pathogenesis and the development of new therapeutic agents and strategies for management in RA have improved clinical outcomes. In addition, the development and application of diverse disease activity measures have contributed to better clinical outcomes through tight monitoring of disease activity and treatment response [3]. In the routine clinical field, the most commonly used indices to evaluate disease activity in RA are Disease Activity Score 28 joints (DAS28) [4], the simplified disease activity index (SDAI) [5], and the clinical disease activity index (CDAI) [6], which are multidimensional instruments that utilize tender and swollen joints, patient and physician global health assessment of disease activity, and acute phase reactants (erythrocyte sedimentation rate [ESR] or C-reactive protein [CRP]).

Complete blood count is a simple, inexpensive, and relatively sensitive clinical index that reflects inflammatory response. Interaction of blood cells including white blood cells, red blood cells, and platelets plays a crucial role in the regulation of inflammation and immune response. Especially, neutrophils, lymphocytes, and platelets are potent effector cells in the inflammatory response [7,8]. Changes in counts of these blood cells in the peripheral blood response to inflammation can be observed in anemia, leukocytosis, and thrombocytopenia [9]. Based on hematological changes in acute phase phenomenon, numerous clinical studies demonstrated that hematological indices such as neutrophil-to-lymphocyte ratio (NLR) or platelet-to-lymphocyte ratio (PLR) were higher in patients with RA compared with controls and adequately reflected disease activity status [10,11,12,13,14,15,16,17]. Recently, the systemic immune-inflammation index (SII), a newly developed marker for systemic inflammatory change, was associated with disease activity in diverse inflammatory rheumatic diseases such as adult-onset Still’s disease (AOSD) [18], antineutrophil cytoplasmic antibody (ANCA)-associated vasculitis [19], and ankylosing spondylitis (AS) [20]. The novel neutrophil-to-hemoglobin and lymphocyte (NHL) score was an independent predictor for a cerebrocardiovascular event [21]. However, it has not been evaluated in the correlation of SII and NHL score with traditional disease activity measures in RA.

Hematological indices are more convenient than disease activity composites commonly used in RA using various clinical and laboratory parameters. Therefore, it is necessary to develop hematological indices that appropriately measure RA activity using a complete blood count. The aim of this study is to determine whether different four hematological indices, i.e., SII and NHL score together with NLR and PLR, can be used to assess disease activity in patients with RA.

## 2. Subjects and Methods

### 2.1. Study Population

This study included 257 female RA patients over the age of 18 years who met the classification criteria for RA as proposed by the American College of Rheumatology (ACR)/European League Against Rheumatism (EULAR) in 2010 [22] and 71 age-matched female controls. RA patients who enrolled in this study were regularly receiving treatment and evaluating disease activity indexes at the Department of Rheumatology in our hospital. In the recruitment of RA patients, patients with other autoimmune diseases, including Sjogren syndrome, systemic lupus erythematosus, mixed connective tissue disease, or systemic sclerosis, were excluded. Control subjects without evidence of diagnosis and treatment for any inflammatory and autoimmune rheumatic diseases were enrolled after a review of medical records and individual interviews at the time of enrollment.

### 2.2. Collection of Clinical Information

Demographic data of age and sex were collected. Disease duration (years), rheumatoid factor (RF, IU/mL), and anti-cyclic citrullinated peptide antibody (anti-CCP antibody, U/mL) were assessed from the review of medical records and the conduction of well-structured interviews with patients. Current anti-rheumatic medications used at the time of enrollment in the study were identified as glucocorticoids, conventional synthetic disease-modifying antirheumatic drugs (csDMARDs; methotrexate, sulfasalazine, hydroxychloroquine, leflunomide, azathioprine, and tacrolimus), targeted synthetic DMARDs (tsDMARDs; tofacitinib and baricitinib), and biological DMARDs (bDMARDs; infliximab, adalimumab, etanercept, golimumab, tocilizumab, and abatacept).

### 2.3. Assessment of Disease Activity Measures in RA

Individual disease activity parameters of swollen joint count (SJC), tender joint count (TJC), patient global assessment (PGA), physician global assessment (PhGA), erythrocyte sedimentation rate (ESR), and C-reactive protein (CRP) were assessed at the time of study enrollment. PGA and PhGA were assessed using a 100 mm visual analogue scale (VAS). Composite disease activity indices used in RA such as DAS28-ESR, DAS28-CRP, CDAI (clinical disease activity index), and SDAI (simplified disease activity index) were assessed. We arbitrarily classified four disease activity categories—remission, low, moderate, and high—into three subgroups based on DAS28-ESR, DAS28-CRP, SDAI, and CDAI to compare hematological indices among three disease activity subgroups, as follows; remission, low, and moderate to high disease.

### 2.4. Assessment of Hematological Indices

Total and differential white blood cells (neutrophils and lymphocytes), hemoglobin, and platelets were assessed from peripheral venous blood obtained from each participant at the time of enrollment in the study. Hematological indices including SII, NHL score, NLR, and PLR were calculated with neutrophils, lymphocytes, hemoglobin, and platelets as follows: SII = platelet × neutrophil/lymphocyte ratio, NHL score = neutrophil/(hemoglobin × lymphocyte) ratio, NLR = neutrophil/lymphocyte ratio, and PLR = platelet/lymphocyte ratio. The unit of NHL score was g/dL.

### 2.5. Statistical Analysis

The data are presented as mean with standard deviation (SD) or median (interquartile range) for continuous variables and number (% of cases) for categorical variables. The Kolmogorov–Smirnov test was used to determine the normality of data distribution. Chi-square test or Fisher’s exact test was performed for the comparison of categorical variables between the two groups. For continuous variables, Student’s *t*-test was used for comparison between the two groups (RA patients and controls) and Kruskal–Wallis tests for comparisons among the three disease activity subgroups (remission, low, and moderate to high disease activity). Pearson’s correlation analysis was used to estimate the correlation between hematological indices and other disease activity variables associated with RA.

Receiver operating characteristic (ROC) curve analysis with estimation of area under the curve (AUC) and 95% confidence interval (CI) was performed to verify values of the hematological indices for diagnosis and remission in RA. The optimal cut-off values of hematological indices for predicting the presence of RA were explored by ROC analysis based on Youden’s index. All statistical analyses were performed using SPSS version 19.0 (SPSS Inc., Chicago, IL, USA). *p* ≤ 0.05 was considered statistically significant.

## 3. Results

### 3.1. General Characteristics of the Study Population

The study population consisted of 257 patients with RA and 71 control subjects. The mean age of RA patients and controls was 60.7 years and 60.4 years, respectively, which was not statistically different (Table 1). All participants were female. The mean disease duration of RA was 11.2 years (SD 8.2 years). The levels and these positivity frequencies of RF and anti-CCP antibody were 99.2 IU/mL (87.9%) and 317.4 U/mL (87.2%), respectively.

Among individual disease activity parameters, ESR and CRR were significantly higher in RA patients than in controls (*p* < 0.001 and *p* = 0.001, respectively). TJC, SJC, PGA, and PhGA were assessed and used for calculation of DAS28-ESR, DAS28-CRP, CDAI, and SDAI. In addition, current medications including glucocorticoid, csDMARDs, bDMARDs, and tsDMARDs for RA were identified.

### 3.2. Comparison of Hematological Indices between RA Patients and Controls

RA patients had higher white blood count, platelet count, absolute neutrophil count, and percentage of neutrophils compared with controls (*p* < 0.001, *p* = 0.002, *p* < 0.001, and *p* < 0.001, respectively) (Table 2). Conversely, RA patients had lower hemoglobin and lymphocyte percentages than controls (*p* = 0.012 and *p* < 0.001, respectively). A difference was not observed in absolute lymphocyte count between RA patients and controls. Regarding hematological indices, RA patients showed significantly higher SII, NHL score, NLR, and PLR than controls (*p* < 0.001, *p* < 0.001 *p* < 0.001, and *p* = 0.046, respectively). There were significant differences in hematological indices between RA and control groups.

### 3.3. Correlation between Hematological Indices and Disease Activity Parameters

The associations between hematological indices and disease activity parameters were evaluated (Table 3). SII was positively associated with ESR, CRP, SJC, PGA, PhGA, DAS28-ESR, DAS28-CRP, CDAI, and SDAI but not with TJC. NHL score was associated with most of all measured activity parameters except for ESR and TJC. NLR was associated with CRP, SJC, PhGA, DAS28-CRP, CDAI, and SDAI. PLR was associated only with CRP but not with other disease parameters.

Four hematological indices were not correlated with titers of RF and anti-CCP antibodies (data not shown). There were no differences in SII, NHL score, NLR, and PLR between RF-positive and -negative patients. The values of all hematological indices were not different according to the presence of anti-CCP antibody. This analysis suggested that hematological indices were generally associated with composite or individual indices of disease activity in patients with RA.

### 3.4. Comparison of Hematological Indices among Disease Activities Based on Composite Measures

Disease activity categories were classified into three subgroups, namely remission, low, and moderate to high activity, based on DAS28-ESR, DAS28-CRP, SDAI, and CDAI. SII, NHL score, and NLR tended to increase gradually as disease activity worsened based on DAS28-ESR, DAS28-CRP, and CDAI, which was statistically significant (Table 4). However, statistical significance was not observed in PLR among disease activity subgroups in DAS28-CRP, SDAI, and CDAI (*p* > 0.05 for all) but was in DAS28-ESR (*p* = 0.021). In addition, no gradually increasing trend in all hematological indices was found among disease activity subgroups based on SDAI, although there was statistical significance among disease activity subgroups.

### 3.5. Assessment of Accuracy of Hematological Indices for the Diagnosis and Remission in RA Using ROC Curve

The ROC analysis was used to determine the diagnostic accuracy of hematological indices in RA. In the ROC analysis, RA was used as the state variable, and hematological indices of SII, NHL score, NLR, and PLR were used as the test variables (Figure 1). The AUC for each hematological index was 0.703 (95% CI 0.637–0.769, *p* < 0.001) for SII, 0.705 (95% CI 0.639–0.770, *p* < 0.001) for NHL score, 0.686 (95% CI 0.619–0.753, *p* < 0.001) for NLR, and 0.581 (95% CI 0.505–0.658, *p* = 0.036) for PLR. The AUC for ESR and CRP was 0.622 (95% CI 0.552–0.693, *p* = 0.002) and 0.660 (95% CI 0.593–0.727, *p* < 0.001), respectively. In addition, the ROC analysis showed that the optimal cut-off values for the presence of RA were 305.6 for SII and 0.112 for NHL score, respectively. Moreover, 85% sensitivity and 42% specificity for SII and 78% sensitivity and 49% specificity for NHL score were yielded based on the Youden’s index. The results showed that SII and NHL score might be statistically acceptable indices for the diagnosis of RA rather than other hematologic indices such as NLR and PLR and acute phase reactants including ESR and CRP.

In addition, the evaluation of efficacy for remission was performed using ROC curves. The AUC obtained from all hematological indices ranged from 0.608 to 0.628, indicating that these hematological indices were too weak to determine clinical remission, although they were statistically significant (Table 5). However, ESR showed significant accuracy for remission but not CRP (AUC = 0.807 for ESR and AUC = 0.692 for CRP, respectively).

## 4. Discussion

Among disease activity measures for RA, acute phase reactants such as ESR and CRP and composite indices including DAS28, SDAI, and CDAI are readily available in clinical practice [3]. The numbers of lymphocytes, platelets, and neutrophils in the peripheral blood or their ratios have been used to provide relevant clinical information for estimating or predicting the systemic inflammatory response. In the present study, the role of hematological indices such as SII, NHL score, NLR, and PLR in the assessment of disease activity in patients with RA was evaluated. The main findings showed hematological indices SII, NHL score, and NLR to be significantly associated with individual disease activity parameters such as ESR or CRP and composite measures. In addition, SII and NHL score, rather than NLR and PLR, provided helpful information for diagnostic efficacy and remission status in RA.

RA is characterized by a chronic inflammatory joint disease combined with joint damage mediated by infiltration of numerous immune and inflammatory cells such as neutrophils, lymphocytes, and macrophages into synovial tissue [1,2]. Neutrophils have emerged as crucial effector cells regulating innate and adaptive immunity. Dysregulated neutrophil activation leads to aberrant inflammatory and immune responses and causes tissue damage through the release of neutrophil-derived pro-inflammatory cytokines and reactive oxygen and nitrogen species in a variety of autoimmune rheumatic diseases such as RA and SLE [23]. In addition, neutrophils facilitate the activation and recruitment of antigen-presenting cells and exhibit a tendency for spontaneous formation of NETs that act as a source of citrullinated autoantigens [24]. Platelets have a pivotal role in homeostasis and coagulation. Recent evidence indicates that platelets are involved in the regulation of inflammation [25]. Inflammatory responses caused by diverse conditions induce activation of platelets in peripheral blood circulation; these platelets then stimulate the release and production of pro-inflammatory cytokines such as interleukin-1. Platelets are important immune effector cells with antigen-presenting properties to increase the immune response of T cells and generate pro-inflammatory microparticles in peripheral blood and synovial fluid [26]. Lymphocytes play a crucial role in the pathogenesis of RA. Infiltration of activated lymphocytes and plasma cells is observed in inflammatory synovial tissue of patients with RA [1,2]. However, lymphopenia in some patients with RA has been identified and associated with severe disease [27]. Although the definite mechanism of lymphopenia has not been determined, decreased numbers of CD4- and CD8-positive T cells or defects in proliferation and differentiation of T cells are presumed relevant [27,28]. Because direct identification and quantification for infiltration of inflammatory or immune cells associated with RA in affected tissues or organs has been limited, indirectly quantifying the distribution of these cells in peripheral blood might be helpful in the assessment of the degree in inflammation. Consequently, the fraction of each blood cell such as SII and NHL score together with NLR or PLR can be used as an inflammatory index depending on the activity or severity of the inflammatory disease.

In the present study, SII and NHL score together with NLR and PLR were used to compare RA activity. SII was calculated using three hematological components of neutrophils, lymphocytes, and platelets, which are considered effector cells involved in the pathogenesis of RA. The NHL score is an index using hemoglobin in addition to lymphocytes and neutrophils. Regarding hemoglobin, inflammation is a common cause of anemia, which is induced by the inhibitory effect of pro-inflammatory cytokines on erythropoiesis in bone marrow and reduced survival of circulating erythrocytes in inflammatory diseases [29]. Anemia is a common comorbidity in patients with RA and associated with more severe or active joint disease [30]. SII has been assessed as a disease activity indicator for several inflammatory rheumatic diseases including AOSD [18], ANCA-associated vasculitis [19], and AS [20]. These studies demonstrated that SII was useful for the assessment of disease activity or as a predictor of poor outcome. Recently, the relationship between SII and disease activity in RA was verified by confirming a statistically significant difference in the non-remission group compared to the remission group based on DAS28 [31]. Based on DAS28-ESR and DAS28-CRP, the present study also confirmed that SII increased statistically as disease activity increased. In contrast, NHL score as an inflammatory hematological marker has been investigated in myocardial infarction [21]. Furthermore, the use of the NHL score for comparing disease activity in RA has not been previously reported. In the present study, SII and NHL score were significantly higher in RA patients than controls and associated with individual disease activity parameters except TJC or ESR and all composite measures including DAS28, SDAI, and CDAI. The poor association between TJC and hematologic indices is compatible with the finding that clinical synovitis reflects radiographic progression at 2 years much better than joint tenderness in RA [32]. In addition, SII and NHL score tended to increase as disease activity based on each composite measure except SDAI increased, indicating that two hematological indices might be useful inflammatory markers for the evaluation of disease activity in RA.

In previous clinical studies, multiple hematological indices such as NLR, PLR, or mean platelet volume have been compared with disease activity in RA [10,11,12,13,14,15,16,17]. RA patients showed significantly higher NLR compared with controls [10,11,12,14,15,16,17], which is in agreement with the finding in the present study. In most studies, a close association between NLR and disease activity indices such as DAS28, ESR, or CRP was reported [11,12,15,16]. Our study found that NLR correlated with CRP, SJC, PhGA, DAS28-CRP, CDAI, and SDAI. Although a close association between NLR and DAS28-ESR was not found, NLR showed a propensity to increase from remission to moderate to high disease activity. Similarly, NLR in active disease was higher than in remission [11,15]. Conversely, NLR was unchanged between active and inactive disease activity in RA [14]. In several longitudinal studies, the change in NLR before and after treatment was evaluated in RA patients. Maden et al. identified a significant decrease in NLR in 82 patients with RA after treatment compared with pretreatment [10]. Treatment with rituximab for 6 months resulted in much lower NLR than pretreatment [12]. In another study, patients with higher NLR treated with TNF blockers showed significantly lower EULAR response at 12 weeks [17]. These results indicate the NLR has potential for monitoring disease activity in RA patients.

In numerous studies about the association between PLR and disease activity in RA, RA patients showed higher PLR than controls [11,12,14,16,17]. In the present study, association was not observed between PLR and all disease activity markers except CRP. Furthermore, PLR did not differ among disease activity groups based on DAS28-CRP, CDAI, and SDAI, but did differ for DAS28-ESR. PLR was increased marginally in RA patients compared with controls (*p* = 0.046). Notably, in contrast to other studies, PLR was not an inflammatory marker in the present study. First, the number of lymphocytes in the PLR analysis plays a decisive role. In the present study, although the fraction of lymphocytes was significantly lower in RA patients compared with controls, a difference was not observed in the absolute lymphocyte count. Yolbas et al. demonstrated that patients with RA had markedly higher PLR values than controls due to differences in lymphocyte count, although the platelet count did not differ between the two groups [14]. Compatible with the present study, PLR did not differ between active disease and remission in early RA [33]. However, several other studies confirmed that PLR was associated with disease activity indices, including DAS28, ESR, and CRP [11,12,16]. In other study, PLR was found to be an available marker that adequately reflects treatment response in RA patients. PLR at baseline was markedly decreased after treatment with rituximab [12]. In addition, EULAR response rate is decreased in RA patients with high PLR [17]. Despite the favorable results regarding the correlation of NLR and PLR with disease activity in patients with RA, a weak correlation was reported in a recent meta-analysis [34], indicating that PLR rather than NLR can be limited as an indicator of disease activity.

It is important to verify that hematological indices have some degree of accuracy in the diagnosis for RA. In the evaluation of diagnostic accuracy for RA, Zhang et al. showed that the combination of NLR-PLR panel had higher accuracy for discriminating active or inactive RA from controls (AUC = 0.880 for active RA and AUC = 0.839 for inactive RA, respectively) [35]. In a multi-center retrospective study, NLR was shown to be a potent hematological index for the diagnosis of RA with an AUC of 0.831 [36]. In the present study, SII or NHL score showed higher diagnostic efficacy for discriminating RA from controls and superior efficacy for diagnosis of RA than NLR or PLR. Based on the study results, hematological indices have higher diagnostic value for RA.

Serologic markers for RA, i.e., positivity or titers of RF and anti-CCP antibody, might be known to be related to disease activity in RA [37,38]. In addition to the association between disease activity and NLR or PLR, one cross-sectional study showed that NLR was significantly associated with RF titer, although there was no difference in NLR between seropositive and seronegative RA patients [14]. Similarly, higher mean NLR and PLR were observed in patients with anti-CCP antibody compared to those without anti-CCP antibody [39]. However, our study found that all hematological indices were not related to titers of RF and anti-CCP antibody and also not different between seropositive and seronegative RA patients.

Until now, various hematological indices have been studied for their relevance to the activity of various rheumatic diseases [10,11,12,13,14,15,16,17,18,19,20]. NLR, PLR, NHL, and SII were introduced and studied as representative hematological indices. These indices have been developed as laboratory indicators that better reflect the activity of individual diseases sequentially by making complex calculation formulas for components such as platelets, neutrophils, lymphocytes, and hemoglobin. However, formulas for hematological indices that can replace the disease activity of each inflammatory disease have not yet been presented, so additional research in this aspect is needed in the future.

There are several limitations in interpreting the results of this study. First, the data from small-sized sample of a single center by a retrospective study design may be a weakness in the robustness of the study results. Since the hematological test results obtained in the retrospective study were measured at separate and different times, the consistency of the results may be lacking. To overcome this limitation, it is necessary to measure collected blood samples in the same laboratory space under the same conditions through a prospective study. Second, the study population consisted of only female patients and healthy controls. There may be differences in the disease activity of RA according to gender [40]. In addition, RA is a representative inflammatory joint disease, and disease control group with non-inflammatory joint disease (e.g., osteoarthritis) is required to verify the efficiency of hematological indices reflecting the degree of inflammation. Third, control subjects participating in this study were selected based on review of medical records and acute phase reactant tests. There was a limitation in not being able to identify the use of drugs or non-inflammatory diseases that could affect the results of hematological indices. Fourth, the individual hematological index did not sufficiently explain the lack of good correlation with remission in RA. This presupposes that the predictive ability of a single index is limited compared to the composite measures. Thus, it is necessary to prove it through a large prospective study in the future in order to further strengthen the hypothesis and results of this study and overcome these limitations.

## 5. Conclusions

In conclusion, hematological indices are easily measured and available disease activity markers of systemic inflammation in RA. Among hematological indices, SII, NHL score, and NLR, but not PLR, were associated with disease activity measures and adequately reflected disease activity in RA. Furthermore, two indices, SII and NHL score, could have a complementary role in the diagnosis of RA. Thus, these hematological indices, along with the traditional measures, are helpful in assessing the disease activity of RA in clinical practice. The role of hematological indices in the assessment of disease activity and prediction of clinical outcome should be confirmed in prospective longitudinal studies.

## Figures and Tables

**Figure 1 medicina-59-00117-f001:**
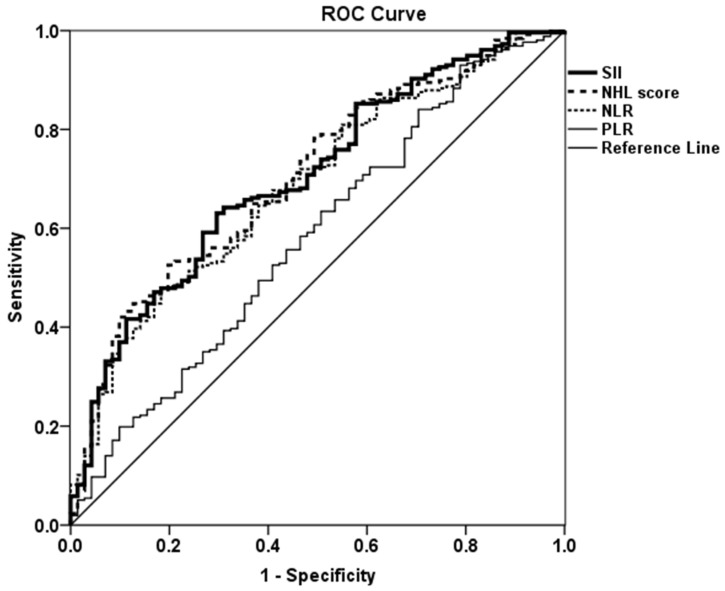
Receiver operating characteristic curve analysis of hematological indices in the diagnosis of RA. Abbreviation: SII, systemic immune-inflammation index; NHL, neutrophil-to-hemoglobin and lymphocyte; NLR, neutrophil-to-lymphocyte ratio; PLR, platelet-to-lymphocyte ratio.

**Table 1 medicina-59-00117-t001:** General characteristics of study population.

Variables	Rheumatoid Arthritis (*n* = 257)	Healthy Controls (*n* = 71)	*p* Value
Age (year)	60.7 ± 8.7	60.4 ± 7.5	0.797
Disease duration (year)	11.2 ± 8.2		
Rheumatoid factor (IU/mL)	99.2 ± 147.9		
Rheumatoid factor (≥14 IU/mL), n (%)	226 (87.9)		
Anti-cyclic citrullinated peptide (U/mL)	317.4 ± 198.9		
Anti-cyclic citrullinated peptide (≥17 U/mL), n (%)	224 (87.2)		
Disease activity indices			
Erythrocyte sedimentation rate (mm/h)	28.5 ± 20.8	19.8 ± 13.5	<0.001
C-reactive protein (mg/L)	5.0 ± 11.5	1.9 ± 4.5	0.001
Tender joint count	3.02 ± 3.27		
Swollen joint count	0.74 ± 2.02		
Patient VAS (mm)	30.2 ± 24.6		
Physician VAS (mm)	19.0 ± 15.3		
DAS28-ESR	3.45 ± 1.22		
DAS28-CRP	2.71 ± 1.12		
CDAI	8.7 ± 7.5		
SDAI	9.2 ± 8.1		
Current medications			
Glucocorticoid	213 (82.9)		
csDMARDs	247 (96.1)		
bDMARDs	79 (30.7)		
tsDMARDs	22 (8.6)		

*p* values were calculated by Student’s *t*-test. Data are described as number of case (%) for qualitative variables or mean (standard deviation, SD) for quantitative variables. Abbreviation: DAS, disease activity score; ESR, erythrocyte sedimentation rate; CRP, C-reactive protein; VAS, visual analogue scale; CDAI, clinical disease activity index; SDAI, simplified disease activity index; csDMARDs, conventional synthetic DMARDs; bDMARDs, biological DMARDs; tsDMARDs, targeted synthetic DMARDs.

**Table 2 medicina-59-00117-t002:** Comparison of hematological indices between rheumatoid arthritis and controls.

Variables	Rheumatoid Arthritis (*n* = 257)	Healthy Controls (*n* = 71)	*p* Value
Whole blood cell counts			
White blood count (×10^3^/µL)	6.9 ± 2.3	5.4 ± 1.3	<0.001
Hemoglobin (g/dL)	12.5 ± 1.2	12.9 ± 1.2	0.012
Platelet (×10^3^/µL)	264.4 ± 73.4	234.1 ± 63.6	0.002
Neutrophil (%)	60.3 ± 10.8	53.3 ± 10.4	<0.001
Lymphocytes (%)	28.9 ± 9.6	36.1 ± 10.0	<0.001
Neutrophil (×10^3^/µL)	4.3 ± 1.9	2.9 ± 1.0	<0.001
Lymphocytes (×10^3^/µL)	1.9 ± 0.7	1.9 ± 0.6	0.736
Hematological indices			
SII	697.1 ± 579.4	409.1 ± 277.4	<0.001
NHL score (/g/dL)	0.207 ± 0.140	0.135 ± 0.080	<0.001
NLR	2.552 ± 1.673	1.693 ± 0.858	<0.001
PLR	158.2 ± 93.6	134.9 ± 55.2	0.046

*p* values were calculated by Student’s *t*-test. Data were described as mean (standard deviation, SD) for quantitative variables. Abbreviation: SII, systemic immune-inflammation index; NHL, neutrophil-to-hemoglobin and lymphocyte; NLR, neutrophil-to-lymphocyte ratio; PLR, platelet-to-lymphocyte ratio.

**Table 3 medicina-59-00117-t003:** Correlation between hematological indices and disease activity indices in rheumatoid arthritis.

Disease Activity Indices	Hematological Indices							
	SII		NHL Score		NLR		PLR	
	*r*	*p*	*r*	*p*	*r*	*p*	*r*	*p*
Composite indices								
DAS28-ESR	0.159	0.011	0.133	0.033	0.111	0.075	0.030	0.637
DAS28-CRP	0.196	0.002	0.183	0.003	0.172	0.006	0.027	0.666
CDAI	0.173	0.006	0.171	0.006	0.158	0.011	0.012	0.844
SDAI	0.144	0.021	0.194	0.002	0.179	0.004	0.031	0.618
Individual indices								
ESR	0.144	0.021	0.107	0.086	0.064	0.307	0.048	0.446
CRP	0.299	<0.001	0.251	<0.001	0.224	<0.001	0.193	0.025
Tender joint count	0.059	0.344	0.048	0.440	0.044	0.485	−0.043	0.493
Swollen joint count	0.215	<0.001	0.230	<0.001	0.214	0.001	0.078	0.215
Physician VAS	0.222	<0.001	0.228	<0.001	0.207	0.001	0.057	0.360
Patient VAS	0.130	0.038	0.124	0.047	0.118	0.059	−0.005	0.939

*p* values were calculated by Pearson correlation analysis. Data were described as correlation coefficient (*r*). Abbreviation: DAS, disease activity score; ESR, erythrocyte sedimentation rate; CRP, C-reactive protein; VAS, visual analogue scale; CDAI, clinical disease activity index; SDAI, simplified disease activity index; SII, systemic immune-inflammation index; NHL, neutrophil-to-hemoglobin and lymphocyte; NLR, neutrophil-to-lymphocyte ratio; PLR, platelet-to-lymphocyte ratio.

**Table 4 medicina-59-00117-t004:** Comparison of the differences in hematological indices among disease activity subgroups based on composite indices.

Hematological Indices	Disease Activity Subgroups Based on Composite Indices			
	Remission	Low	Moderate to High	*p* Value
	DAS28-ESR			
SII	391.2 (265.4–571.6)	543.5 (366.8–801.9)	606.5 (423.5–961.3)	<0.001
NHL score	0.143 (0.102–0.197)	0.188 (0.112–0.238)	0.192 (0.127–0.276)	0.004
NLR	1.771 (1.302–2.545)	2.186 (1.460–3.234)	2.305 (1.618–3.214)	0.021
PLR	124.6 (102.7–156.1)	146.7 (110.4–185.7)	148.4 (119.8–185.9)	0.021
	DAS28-CRP			
SII	440.0 (324.6–733.7)	598.5 (371.7–750.2)	672.5 (423.3–1068.4)	0.001
NHL score	0.150 (0.102–0.225)	0.176 (0.119–0.230)	0.204 (0.131–0.284)	0.005
NLR	1.828 (1.314–2.696)	2.137 (1.485–2.901)	2.531 (1.699–3.322)	0.012
PLR	129.5 (103.1–162.5)	144.4 (116.9–183.6)	147.2 (118.7–186.3)	0.165
	SDAI			
SII	812.5 (505.8–1266.1)	507.6 (342.4–762.4)	757.2 (473.3–1171.3)	0.001
NHL score	0.228 (0.176–0.332)	0.164 (0.115–0.232)	0.232 (0.138–0.300)	0.003
NLR	2.720 (1.910–3.566)	2.001 (1.437–2.864)	2.768 (1.771–3.516)	0.007
PLR	148.1 (121.0–185.6)	137.8 (111.0–173.9)	149.2 (118.7–193.2)	0.293
	CDAI			
SII	433.5 (3.27.1–804.8)	506.0 (338.1–746.1)	759.5 (477.5–1174.0)	0.001
NHL score	0.160 (0.110–0.267)	0.162 (0.110–0.223)	0.232 (0.142–0.303)	0.002
NLR	1.944 (1.384–3.455)	1.954 (1.405–2.727)	2.775 (1.773–3.553)	0.003
PLR	131.4 (107.4–163.5)	137.3 (108.9–172.4)	149.9 (119.0–196.0)	0.314

*p* values were calculated by Kruskal–Wallis test. Data are described as median (interquartile range). Abbreviation: DAS, disease activity score; ESR, erythrocyte sedimentation rate; CRP, C-reactive protein; VAS, visual analogue scale; CDAI, clinical disease activity index; SDAI, simplified disease activity index; SII, systemic immune-inflammation index; NHL, neutrophil-to-hemoglobin and lymphocyte; NLR, neutrophil-to-lymphocyte ratio; PLR, platelet-to-lymphocyte ratio.

**Table 5 medicina-59-00117-t005:** Analysis by receiver operating characteristic curve of hematological indices and acute phase reactants for remission.

Variables	AUC	95% CI	*p* Value
SII	0.662	0.583–0.741	<0.001
NHL score	0.623	0.544–0.701	0.003
NLR	0.608	0.528–0.688	0.010
PLR	0.628	0.550–0.706	0.002
ESR	0.807	0.749–0.864	<0.001
CRP	0.692	0.627–0.757	<0.001

Abbreviation: AUC, receiver operating characteristic; CI, confidence interval; ESR, erythrocyte sedimentation rate; CRP, C-reactive protein; SII, systemic immune-inflammation index; NHL, neutrophil-to-hemoglobin and lymphocyte ratio; NLR, neutrophil-to-lymphocyte ratio; PLR, platelet-to-lymphocyte ratio.

## Data Availability

The data underlying this article will be shared on reasonable request to the corresponding author.

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
