# Peer review of "Association between Novel Hematological Indices and Measures of Disease Activity in Patients with Rheumatoid Arthritis"

_medicina, 2023, doi:10.3390/medicina59010117_

Round 1

Reviewer 1 Report

Manuscript ID medicina-2081463 " Association between novel hematological indices and measures of disease activity in patients with rheumatoid arthritis”

Especially the Systemic Immune-Inflammation Index is a relatively new index. Although other hematological indices have been studied extensively in RA, there is not enough data on SII. In this sense, it can be considered as an original study.

I have a few suggestions:

Abstract: “The study included 257 female RA patients and age-matched 71 female controls”

Results “3.1. General characteristics of the study population: The study population consisted of 257 patients with RA and 70 control subjects” Is the number of healthy controls 70 or 71? This contradiction should be corrected.

Results: “The levels of RF and anti-CCP antibody were 99.2 IU/mL and 317.4 U/mL, respectively.” It may be more appropriate to give the proportion of patients who are positive for RF and anti-CCP instead of the averages.

Table 1: Please give the reference values of RF and anti CCP.

Discussion: Rather than diagnosis, such indices reflect the inflammatory state. In addition to the healthy population, diseased control groups are required as a control group to support this hypothesis. This should be added to the limitations of the study.

This study comparing DAS-28 with Systemic Immune-Inflammation Index in RA should also be discussed. (Satis S. New Inflammatory Marker Associated with Disease Activity in Rheumatoid Arthritis: The Systemic Immune-Inflammation Index. Curr Health Sci J. 2021 Oct-Dec;47(4):553-557. doi: 10.12865/CHSJ.47.04.11. Epub 2021 Dec 31).

Author Response

Dear Editor

Manuscript ID: medicina-2081463

Title: Association between novel hematological indices and measures of disease activity in patients with rheumatoid arthritis

   Thank for the editor and reviewers of the ‘Medicina’ for reviewing our manuscript. We have made some corrections and clarifications in the revised manuscript according to the editor’s or reviewer's comments. You can find out tracing marks for changes in revised manuscript. The changes are summarized below:

Especially the Systemic Immune-Inflammation Index is a relatively new index. Although other hematological indices have been studied extensively in RA, there is not enough data on SII. In this sense, it can be considered as an original study.

I have a few suggestions:

Abstract: “The study included 257 female RA patients and age-matched 71 female controls” Results “3.1. General characteristics of the study population: The study population consisted of 257 patients with RA and 70 control subjects” Is the number of healthy controls 70 or 71? This contradiction should be corrected.

Answer: Thanks for your kind comment. Actually, this study enrolled age-matched 71 female controls. We find out typographical error for “71 female controls”.

Results: “The levels of RF and anti-CCP antibody were 99.2 IU/mL and 317.4 U/mL, respectively.” It may be more appropriate to give the proportion of patients who are positive for RF and anti-CCP instead of the averages.

Answer: Thanks for kind comment. The positivity frequencies of RF and anti-CCP antibody are added and revised at that sentence, as follows “The levels and these positivity frequencies of RF and anti-CCP antibody were 99.2 IU/mL (87.9%) and 317.4 U/mL (87.2%), respectively.”.

Table 1: Please give the reference values of RF and anti CCP.

Answer: Thanks for kind comment. The reference values of RF and anti-CCP antibody are added in table 1.

Discussion: Rather than diagnosis, such indices reflect the inflammatory state. In addition to the healthy population, diseased control groups are required as a control group to support this hypothesis. This should be added to the limitations of the study.

Answer: This comment is very valuable. Therefore, we add and revise the comment to the need for disease control group, as follows “Second, the study population consisted of only female patients and healthy controls. There may be differences in the disease activity of RA according to gender [39]. In addition, RA is a representative inflammatory joint disease, and disease control group with non-inflammatory joint disease (eg, osteoarthritis) is required to verify the efficiency of hematological indices reflecting the degree of inflammation.”.

This study comparing DAS-28 with Systemic Immune-Inflammation Index in RA should also be discussed. (Satis S. New Inflammatory Marker Associated with Disease Activity in Rheumatoid Arthritis: The Systemic Immune-Inflammation Index. Curr Health Sci J. 2021 Oct-Dec;47(4):553-557. doi: 10.12865/CHSJ.47.04.11. Epub 2021 Dec 31).

Answer: Thanks for valuable comment. We add the discussion for close relationship between SII and DAS28 observed in this study, as follows “Recently, the relationship between SII and disease activity in RA was verified by confirming a statistically significant difference in the non-remission group compared to the re-mission group based on DAS28 [31]. Based on DAS28-ESR and DAS28-CRP, present study found that it was also confirmed that SII increased statistically as disease activity increased.”. One reference “Curr Health Sci J. 2021 Oct-Dec;47(4):553-557” is also newly cited.

Additionally, we provide certificate for English editing from eWorldEditing, Inc.

Reviewer 2 Report

In this manuscript, the authors analyzed the patient’s diagnostic data to create hematological indices and re-evaluate the disease activity in patients with rheumatoid arthritis. There are several issues, which need to resolve before considering for publication.

 a) In this new study, disease activity measures in RA. If a traditional parameter could measure the RA disease activity, what is the utility of introducing a new parameter? Please explain in the introduction section.

 b) Are those new indices applicable specifically for RA or those are universal for any hematologic disorder? To address this, a similar analysis should be done in 3 or 4 other different hematological disorders.

c) During Diagnosis, generally different pathological laboratories analyze patient samples, and results of the same sample may vary from lab to lab. There is high chance of having a large number of false positive and/ or false negative data in the pool of study subjects. Therefore, it is essential to analyze separately all collected samples in the same set of laboratory space in a batch. How you did it, please explain in the method section.

 d) Please explains in each of the result section what the analyzed data is signifying.

 e) Please write a concise discussion explaining more about the significance of your new findings.

 f) Please provide some mechanistic explanation of the new indices, which could better estimate disease activity.

 g) The author has no right to use patient diagnostic data for research purposes without the patient's consent. This is a serious violation.

Author Response

Dear Editor

Manuscript ID: medicina-2081463

Title: Association between novel hematological indices and measures of disease activity in patients with rheumatoid arthritis

   Thank for the editor and reviewers of the ‘Medicina’ for reviewing our manuscript. We have made some corrections and clarifications in the revised manuscript according to the editor’s or reviewer's comments. You can find out tracing marks for changes in revised manuscript. The changes are summarized below:

In this manuscript, the authors analyzed the patient’s diagnostic data to create hematological indices and re-evaluate the disease activity in patients with rheumatoid arthritis. There are several issues, which need to resolve before considering for publication.

  1. a) In this new study, disease activity measures in RA. If a traditional parameter could measure the RA disease activity, what is the utility of introducing a new parameter? Please explain in the introduction section.

Answer: Your comment is reasonable to clarify why this study is needed. Therefore, we some sentences as follows, “Complete blood count is a simple, inexpensive, and relatively sensitive clinical index that reflects inflammatory response.” and “Hematological indices are more convenient than disease activity composites commonly used in RA using various clinical and laboratory parameters. Therefore, it is necessary to develop hematological indices that appropriately measure RA activity using complete blood count.” at the part of introduction.

  1. b) Are those new indices applicable specifically for RA or those are universal for any hematologic disorder? To address this, a similar analysis should be done in 3 or 4 other different hematological disorders.

Answer: Thanks for your valuable comment. As described in the introductory part of this manuscript, it has been verified that some hematological indices in various inflammatory rheumatic diseases including ankylosing spondylitis, vasculitis, and adult-onset Still’s disease reflect the disease activity of each disease to some extent. However, please understand that it is difficult to verify many research results on the role of these hematological indices in hematological disorders.

  1. c) During Diagnosis, generally different pathological laboratories analyze patient samples, and results of the same sample may vary from lab to lab. There is high chance of having a large number of false positive and/ or false negative data in the pool of study subjects. Therefore, it is essential to analyze separately all collected samples in the same set of laboratory space in a batch. How you did it, please explain in the method section.

Answer: Thanks for your valuable comment. The reviewer's concerns about error in the analysis of blood samples are fully understood. However, since this study is a retrospective study and blood test findings are obtained from on clinical practice, it is difficult to explain the appropriate method in collecting and analyzing blood samples separately. However, it is judged that the quality control conducted periodically by the hospital is responsible for the consistency of the blood test results. It seems to be a limitation of this study. Therefore, we add detailed description for this limitation as follows, “Since the hematological test results obtained in the retrospective study are measured at separate and different times, the consistency of the results may be lacking. To overcome this limitation, it is necessary to measure collected blood samples in the same laboratory space under the same conditions through a prospective study.”.

  1. d) Please explains in each of the result section what the analyzed data is signifying.

Answer: Thanks for your kind comment. We add the significances of data for 3.2 and 3.3 of the results in this study, as follows “There were significant differences of hematological indices between RA and control groups” and “This analysis suggested that hematological indices were generally associated with composite or individual indices of disease activity in patients with RA.”, respectively. Other subsections of the results including 3.4 and 3.5 already had summarized descriptions for the significances.

  1. e) Please write a concise discussion explaining more about the significance of your new findings.

Answer: Thanks for your comment. according to your comment, we add more description about the relationship between a novel hematological index, SII, and traditional index, DAS28 in the discussion, as follows “Recently, the relationship between SII and disease activity in RA was verified by confirming a statistically significant difference in the non-remission group compared to the re-mission group based on DAS28 [31]. Based on DAS28-ESR and DAS28-CRP, present study found that it was also confirmed that SII increased statistically as disease activity increased.”.

  1. f) Please provide some mechanistic explanation of the new indices, which could better estimate disease activity.

Answer: There is a very obscure aspect of the term “mechanistic explanation”. I'm not sure if this is a sufficient and accurate answer to the reviewer's comment. We add the sentence in the discussion, as follows “Until now, various hematological indices have been studied for their relevance to the activity of various rheumatic diseases [10, 11, 12, 13, 14, 15, 16, 17, 18, 19, 20]. NLR, PLR, NHL, and SII were introduced and studied as representative hematological indices. These indices have been developed as laboratory indicators that better reflect the activity of individual diseases sequentially by making complex calculation formulas for components such as platelets, neutrophils, lymphocytes, and hemoglobin. However, formulas for hematological indices that can replace the disease activity of each inflammatory disease have not yet been presented, so additional research in this aspect is needed in the future.”.

  1. g) The author has no right to use patient diagnostic data for research purposes without the patient's consent. This is a serious violation.

Answer: Thanks for your valuable comment. As in many other retrospective studies, the IRB of our hospital allowed consent waiver to use previous data using patients' medical records. Based on the ethical guidelines in clinical research commonly used throughout South Korea, we are sure that it is not a violation to use previous medical records in this study.

Additionally, we provide certificate for English editing from eWorldEditing, Inc.

Round 2

Reviewer 2 Report

The manuscript has been revised satisfactorily.